# Chitosan/Silver Nanoparticle/Graphene Oxide Nanocomposites with Multi-Drug Release, Antimicrobial, and Photothermal Conversion Functions

**DOI:** 10.3390/ma14092351

**Published:** 2021-04-30

**Authors:** Zheng Su, Daye Sun, Li Zhang, Miaomiao He, Yulin Jiang, Bronagh Millar, Paula Douglas, Davide Mariotti, Paul Maguire, Dan Sun

**Affiliations:** 1Department of Orthopedics, The First Affiliated Hospital of USTC, Division of Life Sciences and Medicine, University of Science and Technology of China, Hefei 230001, China; suz924@mail.ustc.edu.cn; 2School of Mechanical & Aerospace Engineering, Queens University Belfast, Belfast BT9 5AH, UK; dsun03@qub.ac.uk (D.S.); b.millar@qub.ac.uk (B.M.); p.douglas@qub.ac.uk (P.D.); 3Research Center for Nano-Biomaterials, Analytical & Testing Center, Sichuan University, Chengdu 610065, China; zhangli9111@126.com (L.Z.); hemmiao@126.com (M.H.); yljiangss@126.com (Y.J.); 4Nanotechnology and Integrated Bioengineering Center (NIBEC), Ulster University, Co Antrim BT37 OQB, UK; d.mariotti@ulster.ac.uk (D.M.); pd.maguire@ulster.ac.uk (P.M.)

**Keywords:** chitosan, silver nanoparticles, graphene oxide, nanocomposites, antibacterial, drug delivery, photothermal conversion

## Abstract

In this work, we designed and fabricated a multifunctional nanocomposite system that consists of chitosan, raspberry-like silver nanoparticles, and graphene oxide. The room temperature atmospheric pressure microplasma (RT-APM) process provides a rapid, facile, and environmentally-friendly method for introducing silver nanoparticles into the composite system. Our composite can achieve a pH controlled single and/or dual drug release. Under pH 7.4 for methyl blue loaded on chitosan, the drug release profile features a burst release during the first 10 h, followed by a more stabilized release of 70–80% after 40–50 h. For fluorescein sodium loaded on graphene oxide, the drug release only reached 45% towards the end of 240 h. When the composite acted as a dual drug release system, the interaction of fluorescein sodium and methyl blue slowed down the methyl blue release rate. Under pH 4, both single and dual drug systems showed a much higher release rate. In addition, our composite system demonstrated strong antibacterial abilities against *E. coli* and *S. aureus*, as well as an excellent photothermal conversion effect under irradiation of near infrared lasers. The photothermal conversion efficiency can be controlled by the laser power. These unique functionalities of our nanocomposite point to its potential application in multiple areas, such as multimodal therapeutics in healthcare, water treatment, and anti-microbials, among others.

## 1. Introduction

Chitosan (CS) exhibits biocompatibility and biodegradability, and has been explored widely for biomedical applications such as bone/skin regeneration, wound dressing, and cancer therapy [1,2,3]. Graphene oxide (GO), on the other hand, has excellent mechanical properties [4]. Its high specific surface area and oxygen-containing surface functional groups have enabled its application as a drug carrier [5,6]. In addition, GO has also emerged as a promising photothermal agent for photothermal therapy, especially due to its high absorption cross-sections in the near infrared (NIR) region [7,8]. These unique physio-chemical properties have enabled its wide application in the biomedical field, including drug delivery, photothermal therapy, and tissue regeneration [9,10].

Combining CS with GO could lead to a nanocomposite with unique functionality for a wide range of applications, particularly in the biomedical field. For instance, Barra et al. [11] developed electrically conductive CS/reduced GO flexible bio-nanocomposites for food packaging and biological applications. Hermenean et al. [12] fabricated a CS/GO 3D scaffold via pi-pi interaction, and the resulting composite has been proposed for bone regeneration. Rana et al. [13] used CS to surface functionalize GO through chemical crosslinking, with the in-vitro study showing the higher cell viability of the functionalized GO. Bao et al. [14] fabricated biocompatible CS-functionalized GO, which can be used as a nanocarrier for drug and gene delivery. Shamekhi et al. [15] prepared GO containing CS scaffolds for cartilage tissue engineering with enhanced physical and mechanical properties via cross-linking and incorporation of nanoparticles. Liu et al. [16] obtained 3D CS/GO scaffolds with aligned pores for tissue engineering with enhanced mechanical strength, shape-memory, cell scaffolding and protein adsorption ability. GO has also been used to improve the electrical conductivity of biomaterials in cardiac tissue engineering [17,18].

In several recent studies, silver nanoparticles (AgNPs) have been introduced to GO/CS system to endow the nanocomposite with additional antibacterial properties. For instance, Khawaja et al. prepared a GO/CS/AgNPs nanocomposite, where AgNPs were successfully prepared using green tea leaf extracts. The nanocomposite has been demonstrated as a highly effective antibacterial agent against pathogenic strains [19]. Keshvardoostchokami et al. [20] prepared a similar GO/CS/AgNPs nanocomposite for antimicrobial applications, where NaOH was used as the reducing agent for synthesizing AgNPs. Jena et al. [21] also deployed a GO/CS/AgNPs nanocomposite as a coating with anticorrosion and antibacterial properties for marine applications. Despite the promising applications of the GO/CS/AgNPs nanocomposite system in the biomedical field, the photothermal function of GO and the potential of CS as a drug carrier within such a system have not been fully exploited. It is also worth noting that the traditional AgNPs synthesis methods present several limitations such as the use of harsh chemicals, elevated temperature, and/or a long processing time.

The room temperature atmospheric pressure microplasma (RT-APM) process is a rapid, facile, and environmentally-friendly method that can be used to prepare AgNPs. During RT-APM, the solvated electrons and the rich plasma induced liquid chemistry contribute to the nucleation and growth of the AgNPs, and the rapid process (in minutes) eliminates the use of harsh/hazardous chemicals [22,23,24,25]. Previous study also shows that RT-APM synthesized CS/AgNPs nanocomposites have demonstrated tailored structure/properties as well as promising antibacterial activity against *E. coli* and *S. aureus* [26]. One interesting aspect worth further investigation is to introduce GO into the RT-APM synthesized CS/AgNPs, and fully exploit the functionality of each constituent material in this new hybrid CS/AgNPs/GO system for a wider range of functional applications.

In this work, GO was introduced into RT-APM synthesized CS/AgNPs for the first time with the aim of producing a multifunctional chitosan/raspberry-like AgNPs/graphene oxide (CS/RB-AgNPs/GO) composite system. The unique functionality of each constituent material, i.e., the antimicrobial property of AgNPs, the pH responsive behavior of CS, and the photothermal converting property of GO, have been fully exploited for the first time for such a composite system. The novel multi-functionalities demonstrated by our CS/RB-AgNPs/GO composite system could enable its potential application in anti-microbials, drug release platforms, and photothermal cancer therapy, among other areas.

## 2. Materials and Methods

### 2.1. Materials

Chitosan (CS, low molecular weight 50,000–190,000 Da, 75–85% deacetylated), silver nitrate (AgNO_3_, ACS reagent, ≥99.0%), fluorescein sodium, methylene blue, and phosphate buffered saline (PBS) tablets (pH = 7.4) were purchased from Sigma-Aldrich (Gillingham, UK). Graphene oxide (5 mg/mL aqueous solution, flake size: 0.5–5 µm, 1 atomic layer at least 60%) was purchased from Graphene Laboratories, Inc (Calverton, NY, USA). Acetic acid (puriss, glacial, ≥99.9%) was obtained from Riedel-de Haën Sigma-Aldrich Laborchemikalien GmbH (Heidenheim a. d. Brenz, Germany). Ultrapure water (>18.2 MΩ cm Millipore Milli-Q system) was used in all experiments. Helium gas was purchased from BOC, London, UK. Model drugs methylene blue (MB) and fluorescein (FL) were obtained from Sigma-Aldrich, UK.

### 2.2. Fabrication CS/RB-AgNPs/GO Nanocomposite

The room temperature atmospheric pressure microplasma process was used to synthesize CS/AgNPs following the established procedure [25,26]. More specifically, as purchased CS powder was dissolved in 2% (*v*/*v*) acetic acid and stirred for 24 h under room temperature to obtain a 2% (*w*/*v*) homogeneous solution, an appropriate amount of AgNO_3_ (4.0 mM) was added to the CS solution to obtain the precursor solution CS + AgNO_3_ mixture. The mixture was subject to RT-APM treatment and Scheme 1a gives the details of the RT-APM setup. A stainless-steel capillary cathode (0.25 mm inner diameter and 0.5 mm outer diameter) was placed ~1.5 mm above the solution surface, and a carbon rod (anode) was immersed in the liquid. Helium gas was passed through the cathode capillary at a flow rate of 25 SCCM. A constant current of 5 mA and a voltage of 0.7–1.0 kV were maintained during the plasma treatment. The RT-APM treatment was carried out for 10 min, and the samples were stirred during the treatment to obtain a well-mixed colloid (see Scheme 1b). The resulting sample is named CS/AgNPs.

The CS/RB-AgNPs/GO nanocomposite with 0, 0.5, and 1.5 wt% GO were prepared by adding the appropriate amount of GO into the RT-APM treated CS/AgNPs samples followed by 30 min of sonication. All liquid samples were then transferred into a Teflon mold (with 9 mm diameter × 4 mm recess). The samples were frozen overnight at −35 °C, and then by freeze-dried (Lyotrap LTE, lte scientific ltd., Oldham, United Kingdom for 24 h at −60 °C. Table 1 shows the composition and nomenclatures of the samples prepared in this study.

### 2.3. Materials Characterization

A Fourier transform infrared spectrometer (FTIR, Perkin Elmer Spectrum 100 ATR, 4 cm^−1^ resolution and 36 scans per sample) was used to analyze the samples. Thermogravimetric analysis (TGA, TA Instruments SDT-Q600, New Castle, DE, USA) was carried out under nitrogen atmosphere over the temperature range 40–600 °C at 10 °C per min increments. The morphology of the cryo-fractured sample (gold sputtered) was observed by scanning electron microscopy (SEM, HITACHI FlexSEM 1000, Tokyo, Japan) with acceleration voltage in the range of 5–15 kV. Differential scanning calorimetry (DSC) was carried out using a Perkin-Elmer DSC 6 instrument. The samples were first scanned at 20–110 °C at 10 °C/min to remove water and eliminate thermal stress. The samples were then held at 110 °C for 10 min and cooled from 110 °C to 20 °C at a rate of 10 °C/min. The second scan was carried out from 20 °C to 380 °C at 10 °C/min. UV-vis spectroscopy (Cary Win UV, Santa Clara, CA, USA) was used to characterize drug release under pH 7.4 and pH 4, respectively. X-ray Diffraction (XRD, PANalytical X’PERT ProMPD) was used to analyze the crystallinity of samples. Lloyd’s LRX Tensile Tester was used for the quasi-static mechanical testing in compression with a 50 N load cell, following ASTM D1621–10 [27] with specimen sizes (10 mm diameter, 5 mm height). Three measurements were taken for each sample [28].

### 2.4. Drug Releasing Tests

Two model drugs fluorescein sodium (FL) and methyl blue (MB) were selected for single and dual drug release, following the established protocol [29]. To investigate the single drug release profile, FL loaded GO (1.5 mg FL per 100 mg GO), namely (GO-1.5)-FL, or MB loaded CS/AgNPs (0.2 mg MB per mL of CS/AgNP colloid), namely (CS/AgNPs)-MB, was used to prepare the single drug laden nanocomposite, respectively. For dual drug release, both FL loaded GO and MB loaded CS/AgNPs were used. Three types of drug laden nanocomposites were prepared, namely, (CS/RB-AgNPs)-MB/GO-1.5, CS/RB-AgNPs/(GO-1.5)-FL, and (CS/RB-AgNPs)-MB/(GO-1.5)-FL. The drug release test was conducted in 10 mL PBS (pH 7.4 or 4.0, 37 °C). At different time intervals, 2.5 mL of test solution was analyzed using UV-vis spectrophotometer, and the remaining solution was topped up with fresh PBS. Three repeated measurements were performed for each sample and the cumulative release percentage of FL and MB was calculated following Equation (1):Cumulative drug release (%) = M_t_/M_0_ × 100%(1)
where M_t_ is the amount of drug released at time t and M_0_ is the initial amount of drug loaded.

### 2.5. Antibacterial Tests

The antibacterial properties of the nanocomposites were evaluated following established method using gram-negative (*E. coli*) and gram-positive (*S. aureus*) strains [29,30]. 100 μL of the overnight bacterial culture (1 × 10^8^ CFU/mL) was spread on nutrient agar plates. The sample was then placed on the plates in triplicates, followed by incubation for 24 h at 37 °C. After the incubation period, the inhibition length (i.e., the diameter of the inhibition zone surrounding the samples) was measured (*n* = 3), and the average value was used to determine anti-bacterial activity. More details regarding inhibition length measurement can be found in Appendix A.

### 2.6. Statistical Analysis

Statistical analysis was performed using SPSS software (version 22.0; IBM Corp., Armonk, NY, USA), and the statistical significance was analyzed using one-way analysis of variance (ANOVA) followed by the LSD post hoc test. The relationships between different parameters were calculated using Pearson’s correlation test. Statistical significance was considered for *p* < 0.05, while high significance was *p* < 0.01.

### 2.7. Photothermal Conversion Tests

We followed the standard method for photothermal conversion measurements [31]. IR thermal camera (FORTIC 220RD PCBA, Shanghai, China) was used to capture the thermal image of samples under near infrared (NIR) 808 nm laser irradiation with different laser powers (0.2, 0.4, 0.6, and 1.0 W), with a light window of 50.24 mm^2^. The temperature data were analyzed using AnalyzIR software (4.3.1.25).

## 3. Results

### 3.1. Materials Characterization

Figure 1a shows the UV-vis spectra of all testing samples with their optical images shown in Figure 1b. In theory, GO should have an absorbance peak at around 200 nm due to the pi-pi* conjugation of C=C bonds [26]. However, this peak overlaps with the characteristic peak of pure CS at 200–250 nm, and cannot be easily distinguished in the spectra of the CS/RB-AgNPs/GO composite samples. Figure 1b shows the change of solution color for different samples, where brown is indicative of the formation of AgNPs. The finding is consistent with previous studies on RT-APM synthesized AgNPs, where the presence of AgNPs is supported by the typical broad surface plasmon resonance (SPR) absorption peak at around 410 nm [25,26]. The TEM image in Figure 1c shows the morphology and size distribution of RB-AgNPs in CS/RB-AgNPs/GO-1.5 solution with an average size of 73.69 nm (Figure 1d). It is worth noting from Appendix A that in the RT-APM synthesized CS/AgNPs system (i.e., without GO), the AgNPs appear to be more spherical, have smaller particle sizes (mean particle size ~27.5 nm), and are better dispersed. Whereas for CS/RB-AgNPs/GO-1.5, the AgNPs appear to be more agglomerated (raspberry-like) and have a larger particle size (mean particle size ~73.69 nm). It is known that GO has abundant negatively charged oxygen-containing surface functional groups (e.g., –OH and –COOH), whereas CS is positively charged in aqueous solutions due to the protonation of the surface amine group [29]. The strong electrostatic interaction between CS and GO have been reported by many researchers [32,33,34], and such interaction mechanisms have been deployed for layer-by-layer self-assembled CS/GO coatings [29] and nanocapsules [35]. In this study, it is possible that the electrostatic interaction between CS and GO in our CS/RB-AgNPs/GO-1.5 system contributed to the unique morphology formation of RB-AgNP, where multiple AgNPs tend to gather/agglomerate around the sites where CS and GO interact electrostatically, leading to the raspberry-like microstructure of AgNPs.

Figure 2a shows the optical images of pure CS and nanocomposites with different chemical compositions. The TGA results provide a better understanding of the thermal decomposition and thermal stability of pure chitosan and its composites. Figure 2b shows the DSC curves of pure CS, CS/AgNPs, and CS/RB-AgNPs/GO composites obtained from the second heating. The CS chain segmental mobility could be reflected by the glass transition temperature (Tg). The Tg of pure CS is 134.70 °C, and the broad baseline step is due to the rigidity of the CS molecular structure [36]. The presence of AgNPs increased the Tg of CS/AgNPs composites to 138.89 °C, whereas the presence of both GO and RB-AgNPs increased the Tg of CS/RB-AgNPs/GO-1.5 composites to 141.91 °C. The increase in Tg is ascribed to the interaction between CS chains and GO sheets and RB-AgNPs [37,38]. Such interaction may constrain the segmental motion of CS chains by hydrogen bonding and electrostatic interaction and hence increase its thermal stability [39]. The exothermic peak around 270–350 °C shown in Figure 2b and Appendix A indicates the thermal decomposition of composites, which is associated to the material weight loss seen in the TGA curves (Figure 2c). Our pure CS sample shows an initial decomposition of 270.16 °C. The decomposition temperature depends on the polymer molecular weight and may be related to its hydrolytic depolymerization [40]. As shown in Figure 2c, all TGA curves have a similar decomposition trend, and the weight loss around 127.32 °C is attributed to the loss of water from the polymer. The presence of RB-AgNPs and GO delay the initial decomposition of the CS’s polymer chains, and lead to a higher residual mass than pure CS at 700 °C. This can be explained by the decomposition hindering effect [41]. Figure 2d presents the FT-IR spectra of all samples. The characteristic bands of amide I and amide II groups are located at 1654 cm^−1^ and 1385 cm^−1^. The peaks at 1333 cm^−1^ and 1258 cm^−1^ belong to the O-H bending vibration, and the anti-symmetrical stretching of the C-O-C bridge can be found at 1151 cm^−1^ [42,43]. The strong broad band at 3100–3650 cm^−1^ represents the characteristic peak of N-H or O-H stretching vibration from CS, RB-AgNPs or GO [44]. The intensity of the N-H bending vibration peak at 1532 cm^−1^ decreases due to the interaction between AgNPs (or RB-AgNPs) and CS through hydrogen bonding [45].

### 3.2. Microstructural and Mechanical Analysis

Further SEM analysis and mechanical testing (see Figure 3) shows the influence of AgNPs and GO on the composite microstructure and compressive modulus. RT-APM synthesized CS/AgNPs has reduced pore size but greater pore density when compared with pure CS, whereas CS/RB-AgNPs/GO-1.5, containing GO sheets with flake size 0.5–5 µm, shows a more similar structure to pure CS. Figure 3d shows the mean compressive elastic modulus of composite samples. The increased modulus for CS/AgNPs may be attributable to hydrogen bond formation between AgNPs and the polymeric matrix, resulting in restricted motion of the polymer backbone chain [46]. When GO is introduced into the system, AgNPs (which are encapsulated by positively charged CS) may interact with the negatively charged GO flakes through electrostatic interaction [19]. This may lead to agglomeration of the AgNPs and/or GO (also see Appendix A), leading to the raspberry-like AgNP morphology. While well dispersed nanofillers have been reported to enhance the composite mechanical properties, it is also widely recognized that agglomeration of nanofiller materials can cause local stress concentration, thus deteriorating the mechanical properties of the resulting composites [47].

### 3.3. In-Vitro Drug Release Profiles

FL and MB were selected as typical model drugs as their molecular weights are similar to some anti-cancer, anti-inflammatory, and bone regeneration drugs [29]. The successful loading of FL onto GO (GO-FL) can be confirmed by the UV-vis spectroscopy. Figure 4a shows the UV-vis spectra of GO, GO-FL, and the as-prepared FL solutions. The characteristic peaks of FL (238, 459 nm and 491 nm) were blue shifted to 231 nm, 448 nm, and 483 nm in the GO-FL sample. The characteristic peaks of MB at 292 nm and 663 nm remain unchanged after loading MB onto CS/AgNPs. These characteristic peaks were used to investigate the drug release behavior of MB (Figure 4b).

For in-vitro drug release tests, we loaded MB onto CS/AgNPs or loaded FL onto GO to investigate the drug release profile of the different single drugs in the resulting CS/RB-AgNPs/GO-1.5 composite system. Subsequently, we combined MB laden CS/AgNPs and FL laden GO to obtain the dual drug release system. Figure 5 presents the percentage of cumulative drug released under neutral (pH = 7.4) and acidic (pH = 4.0) conditions over 240 h. For single drug release of MB (Figure 5a), a burst release was evident under pH 7.4 in the first 10 h, followed by a more stabilized release of 70–80% after 40–50 h. Under pH 4, however, the release rate was significantly accelerated and the drug release reached 100 % from 80 h. In contrast, the single drug FL (Figure 5b) release shows a time lag behavior. Only 30% FL was released in the first 10 h under pH 7.4 and the cumulative release only increased slowly up to 45% towards the end of 240 h. The lower pH also significantly altered the release profile, which featured a continual increase in cumulative drug release at a much higher rate throughout the measurement period. It is worth mentioning that such pH responsive drug release behavior could be beneficial for on-demand cancer treatment applications, as tumorous sites typically present a more acidic micro-environment [48,49]. Figure 5c,d shows the drug release profile of CS/RB-AgNPs/GO-1.5 when it acted a dual release system. It can be seen from Figure 5c that the addition of FL slowed down the MB release rate for both pH conditions as compared with the single drug (MB) release system. In contrast, with the presence of MB, the FL release rate from CS/RB-AgNPs/GO-1.5 increased slightly under both pH conditions. The complex interplay between different drugs and drug carriers will be further elucidated in Section 3.4.

Most GO based dual drug release systems reported in the literature deploy GO as the sole drug carrier for two types of drugs [50,51,52,53]. For instance, Tran et al. [50] developed a pH sensitive dual anticancer drug (doxorubicin and irinotecan)-loaded GO for cancer therapy. Pei et al. [53] synthesized a pH sensitive dual-drug-loaded PEGylated GO, where cisplatin and doxorubicin were loaded onto GO to facilitate combined cancer chemotherapy. Both reports demonstrated faster drug release under acidic conditions, which has been attributed to the stronger hydrogen-bonding between the drugs’ functional group and GO’s surface functions under neutral conditions than under acidic conditions. In the present study, both CS and GO were deployed as drug carriers. The single drug and dual drug release profiles shown here follow a similar trend to those previously reported in multi-drug laden coating systems consisting of layer-by-layer assembled CS and GO [29]. The greater release rate of both drugs under lower pH can be mostly attributed to the swelling of the chitosan matrix under acidic conditions due to the protonation of its amino/imine groups [54]. The swelling of CS results in a more porous composite structure, and therefore a less hindered drug elution pathway. However, in the present study, the levels of cumulative release in the single/dual drug release models are significantly different from the previously reported coating system. This could be attributed to the vastly different composite microstructures adopted in the two studies, which may significantly affect the interfacial interaction between CS and GO and the potential drug elution pathways.

### 3.4. In-Vitro Drug Release Kinetic Model

In order to obtain a suitable drug release model that best describes the drug release profile, several kinetic models have been considered for the drug release systems involved in this study. These include zero order (cumulative percent drug released versus time), first order (log cumulative percent drug retained versus time), Higuchi (cumulative percent drug released versus√time), and Korsmeyer–Peppas (log cumulative percent drug released versus log time). Curve fitting was carried out following [55], and the data can be found in Appendix A. The modelling accuracy was evaluated by comparing the coefficient of determination (R^2^) values. In the Korsmeyer–Peppas (Fickian diffusion) model, the mechanism of drug release was indicated by the value of the release exponent N [56,57]. Porous structures typically exhibit N < 0.45, and diffusion occurs through water-filled pores and the polymer layer due to the swollen matrix [58,59].

The drug release kinetic analysis results from Appendix A suggest that the Korsmeyer–Peppas model (which shows the highest overall R^2^ values) best describes the predominant release mechanism for both single and dual drug systems (Figure 6). The porous structure of the composite was corroborated by the value of release exponent N < 0.45. When fitting the drug release curves based on the Korsemeyer–Peppas model, [60,61], the cumulative amount of drug release (M_t_) could be described by Equation (2):M_t_ = kt^N^(2)
where N was the release exponent in the Korsmeyer–Peppas model, indicative of the drug release mechanism, t was the time elapsed, and k was defined by Equation (3):k = A√(2C_init_ DC_s_))(3)
where A was the cross-sectional area of the sample, C_init_ was the initial concentration of the drug (MB = 0.2 mg/mL and FL = 1.5 wt%), C_s_ was the solubility (70 mg/mL for MB and 500 mg/mL for FL), and D was the diffusion coefficient of the drug (0.42 × 10^−5^ cm^2^/s for pristine FL and 0.79 × 10^−5^ cm^2^/s for pristine MB) [62].

The true diffusion coefficients (D_K_) of each system following the Korsemeyer–Peppas model were obtained through further optimization towards the objective function. The curve fitting values for D_K_, N and, the corresponding R^2^ values can be found in Table 2.

For the dual drug system, the true diffusion coefficient of MB and FL is different from the ideal diffusion coefficient due to the potential host-guest and guest-guest interactions [63] between GO, MB, and FL (see Figure 7). In our drug laden composites, FL was loaded onto GO via weak pi-pi interaction, which can be described as a weak host-guest (GO-FL) interaction. When MB was introduced into the system, the stronger ionic interaction between MB and GO (host-guest (GO-MB)) drove the replacement of the loosely adsorbed FL with MB to form a more stable structure, leading to a slower MB release but a greater FL release. In addition, the repulsive force between the negatively charged functional groups on FL and GO accelerate the desorption process of FL from GO sheet surface.

### 3.5. Antibacterial Ability

The diameters of the inhibition zones (inhibition length) were used to evaluate the nanocomposite’s antibacterial ability against *E. coli* and *S. aureus* (see Figure 8). It can be seen that pure CS exhibits no visible antibacterial ability. In contrast, samples containing AgNPs and RB-AgNPs showed significantly enhanced antibacterial performance. The presence of Ag can generate reactive oxygen species (ROS) when in contact with the bacterial strains, and these ROS could destabilize the bacteria plasma membrane potential, deplete the levels of intracellular adenosine triphosphate, and eventually lead to the death of bacterial cells [64]. The presence of GO (particularly at 1.5% *w*/*w*), can further enhance antibacterial ability as compared to the CS/AgNPs sample. This could be attributed to the GO’s ability to cause oxidative pressure [65] or bacterial cell membrane damage [66].

### 3.6. Photothermal Conversion

All samples were immersed in PBS solution, and their photothermal conversion behavior under 808 nm laser irradiation were examined, with PBS solution being the control. Figure 9a,b show that under 1 W laser irradiation, samples containing no GO exhibited no photothermal conversion effect. For CS/RB-AgNPs/GO composites, the sample containing greater GO content demonstrated greater photothermal conversion efficiency. Figure 9c shows that by manipulating the laser power, the temperature profile of the selected sample (CS/RB-AgNPs/GO-1.5) can be fine-tuned. When 2 W power was used, the samples were heated to 45 °C within 100 s, which is within the therapeutic window for photothermal therapy applications. Figure 9d further confirms the composite samples have sufficient photothermal conversion stability during the cyclic heating/cooling tests. The photothermal performance of CS/RB-AgNPs/GO-1.5 did not deteriorate. This is consistent with other GO (or graphene) containing composite systems reported in the literature with similar photothermal conversion functions [7,29], demonstrating the strong promise of nanographene materials in on-demand photothermal applications.

## 4. Conclusions

In this study, we fabricated a CS/RB-AgNPs/GO hybrid nanocomposite by incorporating GO into RT-APM synthesized CS/AgNPs. The resulting nanocomposite has then been characterized as potentially having several functional applications including multi-drug release, as an anti-microbial, and photothermal conversion. The drug loading/release behavior and the underlying mechanisms have been studied in detail using both single drug and dual drug models. Our composite system demonstrated pH-sensitive release, and the drug release profile can best be described by the Korsmeyer–Peppas model. The nanocomposite also demonstrated significant anti-microbial properties against *E. coli* and *S. aureus* bacteria strains, as well as photothermal conversion properties. These promising functionalities may lead to the potential application of our nanocomposite in multiple fields, such as multi-therapeutic platforms for the treatment of certain cancers, anti-microbial membranes/coatings, water treatment, and many others.

## Data Availability

Data is contained within the article or Appendix A.

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
