# Peer review of "Chitosan/Silver Nanoparticle/Graphene Oxide Nanocomposites with Multi-Drug Release, Antimicrobial, and Photothermal Conversion Functions"

_materials, 2021, doi:10.3390/ma14092351_

Round 1
Reviewer 1 Report
The authors are describing the design and fabrication of a multifunctional nanocomposite system which consists of chitosan, raspberry-like silver nanoparticles and graphene oxide. The study is adequately executed and described. This manuscript can be considered for publication after addressing some minor issues.
Abstract
- The authors are encouraged to include numeric data to point and emphasize their findings.
Introduction
- Please double check the manuscript for typos.
- The authors should emphasize the novelty of the study.
Materials and Methods
- The numbering of the cells in table 1 is confusing and it should be avoided.
- The different characterization techniques should be described in separated individual sections and not in a single paragraph.
- Please include the respective literature for every technique that has been applied.
Results
- The authors are encouraged to present their results separately following the individual sections of the methods that have previously been described. This is going to make it easier for the reader to follow your workflow and understand the results.
- Please include (where necessary) or emphasize comments that compares your findings to similar systems that already exist in the literature.
Reviewer 2 Report
- The knowledge gap and novelty are not clear in this paper. In the first and second paragraphs of the introduction, you did a literature review; however, there is no knowledge gap or area of improvement identified. There is no clear connection between the review and the proposed solution/idea.
- Some words should be italicized: in-vitro, et al., S. aureus, ... .
- Subscript "t" and "0" in Formula 1.
- How did you measure the zone of inhibition? Clarify it in section 2.5. In Figure 8, what is the inhibition length? Is it diameter?! You never mentioned inhibition length in your text, but you brought it in the Figure 8 y-axis.
- Have you used any statistical analysis? You should include a proper statistical analysis in your paper for comparing the data.
- Line 212: Where is Fig S2? Should it be Fig 2? Same thing with Line 272, 263, and 187.
- Insert a scale bar on Figure 2.a.
Reviewer 3 Report
The authors present an interesting article in which they describe the production of, and subsequent characterisation, of a nanocomposite material with drug-eluting functionality. Specifically, the authors show their chitosan:silver nanoparticle:graphene oxide composite not only facilitates a pH-controlled release of drugs, using functional and traceable stand-ins in methylene blue and fluorescein, but that the material itself exudes anti-bacterial properties, impeding the growth of problematic microbes such as E. Coli and S. Aureus. Taken together, these data indicate there is potential in the material based on the examined properties, with possible applications spanning several fields of interest.
In reviewing the manuscript however I had a number of concerns. The following should be considered when submitting a revised manuscript:
- The writing for the most part is quite good, however in the introduction there are parts where the flow is lacking. Several statements of supporting information are delivered, without much context or relation to the article therein. The authors should revise these instances and improve how the material relates to the subject matter of this article.
- On the topic of writing, there is no discussion to speak of. The manuscript jumps from results to conclusions, without much examination of the results in relation to the existing literature and/or contexts in which these findings might have impact. The authors should consider including a discussion, or modifying the conclusion, to relate the impact of these findings to the existing literature.
- The writing/labels on several of the graphs is quite difficult to read/make out. In particular Figure 1, 2, and 4 could be revised in both the resolution, formatting, and placement of labels to improve this.
- The formatting of the tables needs attention. The tables for some reason have numbering utilised within them, and also the use of heavier lines to distinguish certain rows and columns from one another would be of big help for making the tables easier to read.
- The formatting of the equations could also use attention. There are a number of symbols used which are superfluous to the actual equation itself (i.e. the use of the underscore ‘_’).
- Information on how the ‘zones of inhibition’ were measured would be useful. This is not entirely clear how this was measured, and as there are several ways of doing this, each different in terms of strength and accuracy, details on how the authors performed this should be included.
- There is no mention of replicates or n-numbers really in this article to determine what the error bars and data represent. It would be important to know how reproducible these results are, and no figure legends outline how many replicates or repeats were performed, no do the bars indicate such. Perhaps scatter plots would be useful here? Otherwise the numbers absolutely need to be included at a minimum.
- Similarly, there is no section outlining how the statistical analysis was performed on this data. Between the lack of information on n-number and this the data is hard to attribute a ‘weight’ to. This must be addressed in any resubmission.
Reviewer 4 Report
Authors proposed a paper entitled “Chitosan/Silver Nanoparticle/Graphene Oxide Nanocomposites 1 with Multi-drug Release, Antimicrobial, and Photothermal 2 Conversion Functions “ for the publication in Materials, MDPI.
This work is worth of investigation, it has a quite good scientific soundness.
The use of English needs to be revised properly, according to my opinion.
The paper deserves to be published after major revisions.
First of all, I suggest adding an abbreviation list, due to the high number of acronyms utilized in the manuscript.
I also would like that the authors add more details about How do they assess that their process is environmental friendly? please add a paragraph explaining this, with the addition of proper references.
Here is the list of other issues:
line 18. why “onto”? “loading drugs onto the polymer”
Line 19. check the English here “ with release profile specific to the drugs used.”
Line 20. “ E. coli and S. aureus » should be in italique.
Line 21. “ These unique functionalities point to it’s the potential of nanocomposite system”. check the English here. it should be something like “point to be the…” However, I suggest rephrasing it.
Line 43. I would say “for surface functionalization”.
Line 64. like the comment in Line 20.
Line 72. even if they are well known, these acronyms or abbreviations should be defined clearly.
I think that the aims of this paper should be clarified, not just saying that “GO was introduced into the RT-APM synthesized CS/raspberry-like silver nanoparticles”
Line 111. do the guidelines of this journal require to write this table caption in capital letters?
“Nomenclatures of Samples and Their Preparation Methods.”
Line 112. the title of this paragraph should be clarified. “characterization” of what?
Line 121. “stress, The samples” here a full stop was expected.
line 129. define these drugs in this paragraph before using their abbreviations “drugs FL and MB have”
Line 140. Equation should be written in a clear manner. please check also the guidelines for this Journal.
Figure 1. I do not clearly see the PSD of the samples. could it be possible to remove the embedment and consider it as figure 1c ?
Line 171. “initial decomposition at 270.16 °C due to the thermal degradation of polymer chains”. could you provide references different from [26] for this?
Figure 2 is already small. If authors embed another figure inside, it would be difficult to appreciate it. I understand that it is just a zoom of a specific region of the curves reported in figure 2b.
Figures 5. I do not understand why the drug release profiles do not start from zero concentration.
In figure 5a and figure 5c the plateau points are clearly defined. However, they are not clear in the black and blue curve of figure 5b and 5d. I think that the two compared curves should not necessarily end at the same time. they should end when the final points confirm a clear plateau point.
Table 2. what is the sense of this numeration in this table? moreover, there is an empy cell. please, fill it.
Line 271. where is table S2 ?
Line 278 and Line 281. these equations should be mentioned in the proper section, not in results. here only the parameters and the fitting values of these equations can be commented and shown.
Figure 6. I suggest leaving only the raw points and the fitting curves in these diagrams. the line-point curves are confusing the diagrams in my opinion. I also think that R-square values should be shown after the fitting curves description.
Figure 8. “24 h” could be indicated in the caption of the figure, not in the legend.
Line 327. “excellent” maybe “efficient”?
Line 343. “Our composite system demonstrated pH sensitive release and the drug release profile can be best described by the Korsmeyer-Peppas model.” did you support this affirmation with a proper number of references?
Round 2
Reviewer 2 Report
- The authors addressed the comments and they revised the manuscript based on the feedback.
- Figures S3 and S4 in SI have not been referred to, in the manuscript.
- The order of authors' names has changed: third and fourth names; and there is an extra "," in there. Remove the comma. The name orders are not compatible with the current submission portal name order.
Author Response
|
Reviewer #2 The authors addressed the comments and they revised the manuscript based on the feedback.
|
|
|
1) Figures S3 and S4 in SI have not been referred to, in the manuscript.
|
These figures (now Fig S4 and S5 in the revision) are referred in the main manuscript, see line 333-334. |
|
2) The order of authors' names has changed: third and fourth names; and there is an extra "," in there. Remove the comma. The name orders are not compatible with the current submission portal name order. |
The order of authors names are now consistent with the submission portal name order. |
Reviewer 3 Report
The authors have responded positively to my comments for the most part, and as a result the manuscript is much improved.
However, some of my points remains partially not addressed.
- The labelling on some of the figures is still difficult to read. For example, Figure 4 is much improved having the labels fall over the picture, whilst there is difficulty reading the labelling in Figure 1 in which the label falls on the picture. This is not the only example, and the reviewers would be advised to review all figures regarding this.
- The methodology for how the zone of inhibition was measured is still lacking. Details on how this was measured are needed.
- While some improvements to the writing have been made, I still feel there is scope to include more discussion in relating the findings to the existing literature.
Author Response
|
Reviewer #3 The authors have responded positively to my comments for the most part, and as a result the manuscript is much improved. |
|
|
1) The labelling on some of the figures is still difficult to read. For example, Figure 4 is much improved having the labels fall over the picture, whilst there is difficulty reading the labelling in Figure 1 in which the label falls on the picture. This is not the only example, and the reviewers would be advised to review all figures regarding this. |
Fig.1 has been revised. Fig 3 has also been revised with in-picture labels removed as these have been clearly explained in the caption. |
|
2) The methodology for how the zone of inhibition was measured is still lacking. Details on how this was measured are needed. |
We provided supplemented details in line 170-171 as well as in Fig S3. |
|
3) While some improvements to the writing have been made, I still feel there is scope to include more discussion in relating the findings to the existing literature |
Discussion has been further expanded with new references included. See highlighted text in Sections 3.1, 3.2 and 3.3. |
Reviewer 4 Report
The authors provided a new version of the paper, that now deserves to be published after addressing the following considerations:
Figure 5
“The reviewer is correct that for Fig 5b and 5d, the curves have not reached plateu and it may take much longer time to reach the stabilized region. We chose to show the first 240 h of data for all cases, so that the drug release behavior of different systems can be compared in parallel within the same time frame.”
I can understand that authors decided to cut the diagrams at 240 h. However, this should be clearly indicated in the manuscript and explained.
However, why figures 6 do not contain explicitly 240 h as final value of the x axis?
Line 327. “The authors do not think “efficient” is the right word to describe “thermal stability”.
I respect your point. However, I think neither “excellent” is the right word. Please use another one.
Author Response
Dear editor,
The authors would like to thank the editor and all the reviewers for their additional comments, which are helpful for the improvement of this manuscript. The authors of this paper have revised the manuscript to address all the comments raised. The additional text has been highlighted in green.
The author would also like to bring to the editor’s attention that the “Author contributions” statement in the submission portal was not accurate. We would like to provide the updated text for the Authors Contributions as follows:
“ZS and Dan Sun conceived the idea of the work. ZS and Daye Sun carried out the experiment, LZ, MH and YJ contributed to the antibacterial tests, BM and PD contributed to the thermal analysis, DM and PM contributed to the room temperature atmospheric pressure plasma instrumentation, Dan Sun is responsible for resources and funding administration.”
Once again, we would like to thank you for your kind consideration for our manuscript to be published in Materials.
Best regards,
Dr. Dan Sun (d.sun@qub.ac.uk )
|
Reviewer #4
|
|
|
1) Figure 5 I can understand that authors decided to cut the diagrams at 240 h. However, this should be clearly indicated in the manuscript and explained. However, why figures 6 do not contain explicitly 240 h as final value of the x axis?
|
Fig 6 has been revised to have x-axis consistent with Fig 5. |
|
2) Line 327. “The authors do not think “efficient” is the right word to describe “thermal stability”. I respect your point. However, I think neither “excellent” is the right word. Please use another one.
|
The word in question has been changed to “sufficient”, see section 3.6 highlight. |